# The Features of Children with Juvenile Idiopathic Arthritis with Cervical Spine Involvement in the Data from a Retrospective Study Cohort

**DOI:** 10.3390/jfmk10010068

**Published:** 2025-02-15

**Authors:** Lubov S. Sorokina, Artem K. Artamonov, Maria A. Kaneva, Natalia A. Gordeeva, Rinat K. Raupov, Alexander Yu. Mushkin, Dmitri O. Ivanov, Mikhail M. Kostik

**Affiliations:** 1Hospital Pediatry Department, Saint Petersburg State Pediatric Medical University, Saint-Petersburg 194100, Russia; lubov.s.sorokina@gmail.com (L.S.S.); artem.artamonov.96@bk.ru (A.K.A.); spb@gpmu.org (D.O.I.); 2Department of Consulting and Diagnostic, Saint-Petersburg Children’s Hospital #2, n.a. Saint Mary Magdalene, Saint-Petersburg 199004, Russia; 3Pediatric Rheumatology Department, H. Turner National Medical Research Center for Children’s Orthopedics and Trauma Surgery, Saint-Petersburg 197136, Russia; rinatraup94@gmail.com; 4Pediatric Orthopedic and Surgery Department, Saint-Petersburg Research Institute of Phthisiopulmonology, Saint-Petersburg 191036, Russia; aymushkin@mail.ru

**Keywords:** juvenile idiopathic arthritis, JIA, cervical spine arthritis, arthritis, biologic, remission

## Abstract

**Background/Objectives:** Cervical spine arthritis (CSA) in children with juvenile idiopathic arthritis (JIA) can lead to clinically significant and irreversible functional impairment. Our study aimed to evaluate the features of the JIA disease course in children with CSA. **Methods:** In the retrospective cohort study, the data from medical charts of children with JIA (n = 753) who corresponded to the ILAR criteria and were treated from 2007 to 2016 were included. CSA was diagnosed by clinical manifestations (pain and limited range of motion) with radiological confirmation in the available cases. **Results:** CSA had 101 JIA patients (13.4%), predominantly with polyarticular (48%, OR = 1.8 (1.2; 2.7), *p* < 0.001) and systemic (18.9%, OR = 3.6 [2.0; 6.6], *p* < 0.001) JIA categories. CSA was associated with longer disease duration, higher inflammatory activity, a higher number of active joints, a lower probability of achieving remission (HR = 1.33 (95% CI: 1.01; 1.76, *p* = 0.04)), and a higher probability of being treated with biologics (HR = 1.78 (95% CI: 1.22; 2.59, *p* = 0.002)). Patients with temporomandibular arthritis (OR = 10.4 [5.4; 19.8], *p* < 0.001) and shoulder arthritis (OR = 14.1 [7.5; 26.3], *p* < 0.001) had the highest risk of having CSA. **Conclusions:** CSA was an independent predictor of treatment with biologics and failure to achieve remission. Identified predictors can help to find the group of patients with higher suspicion for whom the functional tests and MRI are required to not miss the CSA. A radiology assessment of CSA should be performed as far as possible in children, unless there are risks of general anesthesia for younger patients.

## 1. Introduction

Juvenile idiopathic arthritis (JIA) is a chronic inflammatory arthropathy lasting more than 6 weeks in children under 16 years of age when other causes of arthritis are excluded [1]. According to the ILAR classification, JIA consists of the oligoarticular, polyarticular (seropositive and seronegative for rheumatoid factor), enthesitis-associated, systemic, psoriatic, and undifferentiated JIA categories [1]. Cervical spine arthritis (CSA), which is arthritis of the atlantooccipital and atlantoaxial joints, is most common in patients with polyarticular (42%), systemic (16%), and enthesitis-associated (28%) JIA categories and is associated with [2,3,4].

Anatomical features and a lack of physical examination make the diagnostics of CSA difficult. CSA does not present typical signs of peripheral arthritis such as swelling, local hyperemia, and hyperthermia, and it includes neck pain, limited range of motion, morning stiffness in the neck, torticollis, headaches, and neurological disorders in cases of compression of the cervical spine. Limited range of motion is often a more frequent sign of CSA than pain. A lack of routinely checking the range of motion in the CS, even in the absence of clinical complaints, is the most common cause of misdiagnosis of CSA. Limited range of motion is the primary symptom of functional assessment of the CS and is defined as follows: a rotation of less than 90° in rotation and/or less than 45° in active or passive flexion. The absence of clinical symptoms does not exclude subclinical inflammation of CS joints. The radiological findings confirming CSA were detected in 51% of patients without local clinical manifestations [5,6]. The main imaging methods for the structural and inflammatory changes in the CS are magnetic resonance imaging (MRI), computed tomography (CT), and, rarely, X-rays [7]. X-ray only allows for the detection of impressive post-inflammatory changes in CSA and usually is not used in the routing assessment [8]. MRI is the most sensitive tool for the diagnosing of CSA, allowing for the detection of synovitis, spondylitis, hyperplasia of the synovial membrane (pannus), joint effusion, bone marrow edema, the development of enthesophytes and syndesmophytes, and compression of the spinal cord and its radixes [9]. MRI allows for detecting the CSA at the earliest stages of the disease, and their frequency in patients with JIA accounts for 57% [5,6]. MRI is the most common approach for the diagnosis of CSA, and the main limitations in children and adolescents are related to the difficulty in distinguishing between physiological and pathological changes in the synovial membrane of joints since, in children, the signal from a healthy synovial membrane can be amplified in normal. Often, MRI requires sedation in young children and it could limit the application of the method [10]. Early detection of CSA is important for children with JIA because, usually, it requires more aggressive treatment. Delayed diagnostics and ineffective treatment could lead to disability-inducing irreversible changes, such as subluxations, instability, and ankylosis of the CS joints [11].

The last two decades in pediatric rheumatology were marked by a breakthrough in the therapy of JIA associated with the widespread use of biological drugs, which allows for reaching the inactive phase of the disease in a short time and, as a consequence, preventing the development of irreversible changes, including the CS joints. Being a marker of the severity of JIA, the CSA should be properly diagnosed in its early stages. To describe the features of JIA with CSA and factors associated with CSA, we provided the present study.

## 2. Methods and Materials

We extracted the data from the medical histories of patients with JIA (n = 753) treated in 2007–2016 in the retrospective cohort study. The diagnosis of JIA was established according to the ILAR criteria [1].

### 2.1. Assessment and the Outcomes

In every patient, the following data were extracted:(i)Demography: sex, onset age, age of the study inclusion, JIA category according to the ILAR classification [1], presence of the uveitis.(ii)Clinical data: Joints assessment, active joints number. The joint was considered active if it was swollen or if there was pain and restricted movement. Arthritis of the temporo-mandibular joints (TMJs) was considered if the patient had two or more of the following clinical signs: pain in the TMJs, jaw opening limitations, jaw opening deviations, micrognathia, and other orofacial deformities related to JIA involvement.(iii)The presence of antinuclear antibodies (ANA) and the HLA B27 antigen, an erythrocyte sedimentation rate (ESR), and C-reactive protein (CRP).(iv)Treatment: Systemic corticosteroids (oral and intravenous), non-biologic (nb), and biologic (b) disease-modifying anti-rheumatic drugs (DMARDs)(v)Outcomes: The achievement of remission, the presence of the subsequent flare, and time to JIA remission and JIA flare.

Patients were assigned to the group of CSA if the patient corresponded to at least one of the following criteria: (i) CSA was documented in the patient’s medical documents and/or (ii) the patient had clinical manifestations such as morning stiffness, neck pain, limited range of motion in the neck, and torticollis with or without the relevant imaging data.

The study flowchart is depicted in Figure 1.

The imaging assessment was prescribed by the attending physician in the presence of symptoms indicating possible CSA with the consent of the child and/or his/her legal representatives; it was also dependent on the results of the assessment of the feasibility of the study and risk/benefit assessment in cases when sedation of the patient was required for the study.

### 2.2. Ethics

Written consent was obtained according to the Declaration of Helsinki. The local Ethical Committee of Saint Petersburg State Pediatric Medical University (protocol number 11/10 from 23 November 2020) approved this retrospective study’s protocol. All data were included in an anonymized form. Each case history had informed consent signed by the legal representatives to allow for the use of the medical data in the research on an anonymous basis.

### 2.3. Statistical Analysis

The sample size was not calculated initially. A statistical analysis was performed with the software STATISTICA, version 10.0 (StatSoft Inc., Tulsa, OK, USA). All continuous variables were checked by the Kolmogorov–Smirnov test, with no normal distribution being identified. The quantitative variables were median (Me) and percentiles (25%, 75%) for continuous variables and absolute frequencies and percentages for categorical variables. A Pearson’s χ2 test or Fisher’s exact test in the expected frequencies < 5 was used to compare the categorical variables. Two quantitative variables were compared using the Mann–Whitney test. The ability of each variable to discriminate patients with CSA from patients without it was evaluated with a sensitivity and specificity analysis, an AUC-ROC (area under the receiver operating characteristic curve) with a 95% confidence interval (CI), and a calculating odds ratio (OR) for the detection of the best cut-offs of continuous variables. A survival analysis in each group, with JIA outcomes (treatment with biologics, achievement of the remission) as the event of interest, was conducted through the Kaplan–Meier method. The log-rank test compared survival curves. Factors significantly associated with the time of JIA outcomes were then tested in a Cox proportional hazards regression model, calculating the hazard ratio (HR) with a 95% confidence interval (CI). A *p*-value < 0.05 was considered to be statistically significant.

## 3. Results

### 3.1. The Phenotype of Patients with JIA Who Had CSA

CSA was detected in 101 patients (13.4%) and predominantly was diagnosed in patients with polyarticular (48%) and systemic (18.9%) categories of JIA. A significant association between CSA and arthritis of the temporomandibular joints (TMJs) and arthritis of the joints of the upper extremities, hips, and knees was revealed. Patients with CSA had a longer duration of the disease and time to remission, higher laboratory inflammatory activity, and more often received systemic corticosteroids and biologics compared to patients without evidence of CSA. Patients with CSA rarely had chronic anterior uveitis. In both groups, the time from JIA onset to biologic therapy was comparable (2.6–2.8 years), but the proportion of patients receiving biologic therapy was higher in the CSA group. Data are summarized in Table 1.

### 3.2. Factors Associated with CSA

The variables with significant differences, obtained in the univariate analysis, were assessed as the possible predictors of having CSA along with the analysis of the sensitivity (Se), specificity (Sp), and odds ratio (OR) calculation, a Kaplan–Meier survival analysis, and Cox proportional hazards regression methods. Arthritis of the definitive joints was associated with CSA in JIA patients. The data regarding the sensitivity, specificity, and odds ratio of these predictors are in Table 2.

In both groups, the time from JIA onset to biologic therapy was comparable (2.6–2.8 years), but the proportion of patients receiving biologic therapy was higher in the CSA group. CSA increased the likelihood of biologic treatment—HR = 1.78 (95% CI: 1.22; 2.59, *p* = 0.002) and also was an independent predictor of decreased cumulative probability of achieving remission (HR = 1.33 [95% CI: 1.01; 1.76], *p* = 0.04), indicating the severity of this variant of the course of JIA (Figure 2A,B).

Arthritis of the definitive joints was associated with CSA in JIA patients. The data of the sensitivity, specificity, and odds ratio of these predictors are in Table 2.

## 4. Discussion

CSA in JIA patients is not rare but is still an under-recognized problem. The main predictors of CSA were polyarticular and systemic JIA and arthritis of the specific joints, higher disease duration, higher inflammatory activity, a larger number of active joints, the frequency of corticosteroids and biologics, and failure to achieve remission.

CSA was found in 13.4% of our children. Active arthritis, confirmed by MRI, was found in 46% of children, and post-inflammatory changes featured in 39% of patients for whom an MRI was conducted. The frequency of CSA in our study was higher compared to the Ključevšek study (involvement of the CS was based on clinical and MR signs in children having arthritis for more than 3 weeks), at 4% [11]. In another study, 11% of 206 children had clinical signs of CS involvement, and 6% had MRI-confirmed arthritis [12].

X-ray diagnostics has low sensitivity in children with CSA, as it could only demonstrate structural changes typical of the advanced stage of the inflammatory process or its consequences (deformation of the axial vertebral tooth, erosions), dislocations, and ankyloses, which, in this age group, are very rarely the first signs of CSA [13].

“The gold standard of diagnosis of CSA” is contrast-enhanced MRI of the CS [14,15]. It is necessary to note that normal synovial tissue in childhood may have a slight signal enhancement, which makes it difficult to differentiate from the early stages of inflammation. The postcontrast MR image should be obtained within the first 5–10 min after contrast injection, as later assessment of the imaging leads to diffusion of contrast into the joint cavity, which limits differentiation between signal enhancement from the thickened synovial tissue and joint fluid [16]. It should be noted that MR signs and their severity may not correspond to clinical manifestations of the CSA. Thus, in a retrospective study of 43 JIA patients, MRI findings were detected in 28 (65.1%) cases, while one quarter of the patients had no clinical signs of CSA [17]. The authors suggested that a cervical spine MRI should be performed for all patients in the polyarticular JIA category at the onset of the disease as well as at the onset of clinical signs of CSA [17].

### 4.1. The Most Typical CS Changes in Patients with JIA Are as Follows

Ankylosis of the articular joints. Usually, these changes are observed in C2–C3. Ankylosis can result in impaired vertebral body growth, especially in patients with early-onset JIA [3].Atlantoaxial instability. Anterior subluxation of the atlas is characterized by an increase in the distance between the anterior C1 semicircle and the dens of C2. The anterior atlantodental interval (AADI) is the horizontal distance between the posterior cortex of the anterior arch of the atlas (C1) and the anterior cortex of the dens in the median (midsagittal) plane.

An AADI value of more than 5 mm in children indicates mechanical instability or subluxation [18,19]. In older patients, the maximum possible AADI value is 3 mm. This difference is due to physiological hypermobility in children and incomplete vertebral ossification. Quite often, AADI values in children with JIA exceed 3–5 mm, but this is not accompanied by damage to the atlas transverse ligament. In these patients, it is more informative to measure the posterior atlantodental interval (PADI) as the distance between the posterior margin of the dens and the anterior rim of the posterior arch of C1. PADI values greater than 13 mm are safe. A decrease in this value indicates a high risk of neurological impairment. Spinal cord compression due to atlantoaxial instability can lead to severe and life-threatening consequences, such as tetraparesis and sudden death. The most common clinical manifestations are spasticity of the lower limb muscles, mixed tetraparesis, and weakness and atrophy of the hand muscles [3].

3.Atlantoaxial rotational locking. Damage to the joint capsule or ligament apparatus of the atlantoaxial joint can lead to the development of rotational subluxation of C1. With prolonged dislocation, the capsule and ligaments become tightened, which leads to chronic atlantoaxial block. A typical clinical manifestation of a rotational subluxation is a malposition of the head with a slight (about 20°) tilt to one side and rotation to the opposite side. Risk factors for the possible transition of subluxation into chronic block include the contraction of the joint capsule, intra-articular fibrous inclusions, synovitis of the adjacent joint surfaces, formation of C1 and C2 vertebrae bony fusions, and secondary deformity of the surface of the facet joints [20].

There are four types of rotatory subluxation of C1–C2:

First type—An isolated rotation of C1 anterior displacement;

Second type—An anterior displacement not exceeding 5 mm is present;

Third type—An anterior displacement of C1 exceeding 5 mm;

Fourth type—A rotational component combined with a posterior displacement of C1.

Treatment of patients with the third and fourth types of rotatory subluxation required open repositioning, laminectomy, and occipitospondylodesis with spinal metal devices [20].

Occipitoatlantoaxial rotational locking is a rare situation in which the atlas appears displaced in the horizontal plane not only about the C2 vertebra but also to the occipital bone. Rotational subluxation of the atlas usually precedes the development of dislocation of the occipital bone, which serves as a kind of compensatory mechanism to give the patient’s head a neutral position in conditions of chronic atlantoaxial blockage [3,21].

4.Disturbance of the growth and development processes. When CSA occurs, vertebral growth delay is observed in 16–28% of patients. They may be caused not only by inflammation but also by corticosteroid therapy [21]. The MRI of cervical spine arthritis is in Figure 3.

Several studies have shown that the presence of rheumatoid factor, as well as other immunologic markers, does not increase the risk of CSA [2,6]. In our study, CSA more frequently occurred in polyarticular and systemic JIA categories and with arthritis of the joints anatomically closest to the CS: the TMJs and the shoulder [22]. The literature describes that 2% of children with the systemic variant of JIA have CSA at the onset of the disease [23]. Combined involvement of the CS joints and TMJs has also been described previously [1,12,22]. It is interesting that both CSA and TMJ arthritis often have the same clinical presentation—headache and cervicalgia [24]. This symptom is related to muscles, particularly opening the mouth and the muscles of the neck [25,26]. All these muscles attach to the skull. The TMJs and cervical spine form a functional complex named the “cranio-cervico-mandibular system” [27] The convergence of sensory signals from the neck region onto the trigeminal motor neurons triggers an increase in the activity of the muscles involved in chewing, leading to a reflexive contraction of these muscles in response to the contraction of the neck muscles [27,28]. CSA and TMJ arthritis have the same set of predictors: polyarticular and systemic JIA categories, shoulder, elbow, wrist, hip involvement, treatment with oral and high-dose systemic corticosteroids, requirement of biologic, lower probability of achievement of the remission [22]. Both forms of arthritis are predictors of the severity of the JIA and poorer outcomes [22,29,30,31]. Neck muscle spasms are considered one of the main reasons for temporomandibular dysfunction, which closely resembles TMJ arthritis. Vice versa, TMD, often presented as cervicalgia, is required to exclude the arthritis of both joints, the TMJs, and CSA [26]. Patients with TMD had higher angular deviation in cervical joint position sense during flexion and extension (*p* < 0.001). The correlation between pain (VAS-cervical and VAS-TMJ) and the parameters of the functionality and disability (Fonseca Anamnestic Index and Neck Disability Index) of the cervical joints and TMJs was observed [32]. An eight-week neck exercise program improved the TMD outcomes compared to placebo treatment in a randomized controlled trial in 54 women (18–45 ages) with myofascial or mixed TMD [33]. Taking this into account, it is recommended to include the assessment of the TMJs during CS MRI in JIA patients with suspected CS involvement and vice versa in JIA children with possible TMJ arthritis, including CS in the scanning zone [1,12,22].

The incidence of CSA in JIA patients has decreased in recent decades compared to the biologic era (20th century to 21st century), where it was about 60% in patients with JIA with a predominantly polyarticular category [34]. Adult patients with JIA have a higher incidence of CSA, ranging from 30 to 60%, than adult patients with rheumatoid arthritis, which is most likely due to a longer duration of persistent uncontrolled inflammation [6,13,35]. Bone marrow edema of the cervical vertebra might be a predictor of future erosions and is considered a pre-erosive stage of the inflammatory process. Therefore, bone marrow edema can be considered as an indication for the possible escalation of therapy, including the biological administration preventing destructive changes [13,17,36]. According to the 2019 American College of Rheumatology clinical guidelines, arthritis of the axial skeleton could be considered as an indication for biologic treatment in the first line with or without methotrexate, omitting methotrexate therapy in specific cases [29]. The follow-up MRI assessment after initiation of the treatment should be performed after 9 months, and, if therapy is effective, the next study should be performed after 1 year [5,12].

### 4.2. Surgical Treatment for CSA in JIA Patients

In some patients, the surgical treatment should accompany the treatment with biologics or methotrexate. The main indication for surgical interventions is spinal instability complicated by spinal cord compression. According to prevalent clinical complaints and radiological data, the following types of surgery could be applied:

Instrumental stabilization of the craniovertebral zone in adults to reduce the size of the retro-dental pannus in conditions of severe atlantoaxial instability (AAI) [37]

The development of signs of brainstem compression in juvenile chronic arthritis is a direct indication for surgical treatment, primarily of a decompressive-stabilizing nature, regardless of the patient’s age [3].

Apparently, unlike in adults, these interventions should be considered in children as possible emergency options in the treatment of progressive CSA with spinal cord compression or at higher risk of it. The rare indication for spinal instrumentation in JIA patients even without cord compression could be CSA complicated by spinal instability with severe neck pain which is not resolved by medications and collars.

Future studies might be addressed to assess the functionality of the CS as a surrogate biomarker of disease activity and articular damage in patients with JIA and CSA.

The main limitations of our study are related to the small sample size, retrospective study design, absence of indication to MRI assessment, partial MRI coverage related to young age and necessary general anesthesia, a lack of awareness of physicians against CSA, and selection of the patients with the most prominent clinical signs. Different disease duration might be a factor in different incidences of the CSA in our cohort. We understand that all the abovementioned limitations could influence the study results.

## 5. Conclusions

CSA is a factor of unfavorable prognosis and requires timely and adequate therapy, while progress in the treatment of children with JIA will reduce the frequency of gross post-inflammatory, disabling changes not only in peripheral but also in axial joints, including the joints of the CS.

The involvement of CS in patients with JIA is a serious diagnostic and therapeutic problem. The absence of characteristic clinical manifestations and the possibility of an asymptomatic or asymptomatic course of the inflammatory process requires a targeted search for indications for contrast-enhanced MRI. Assessment of the head and neck range of motion, especially rotational motions, should be performed for patients with JIA even in the absence of the subject complaining, especially in children with polyarticular and systemic JIA categories, and in children with TMJ and shoulder arthritis. The absence of standardized indications for MRI in children with JIA with suspected CSA required setting up this diagnostic protocol based on the abovementioned risk factors. Early diagnosis of CSA help to avoid severe damage and life-threatening complications.

## Figures and Tables

**Figure 1 jfmk-10-00068-f001:**
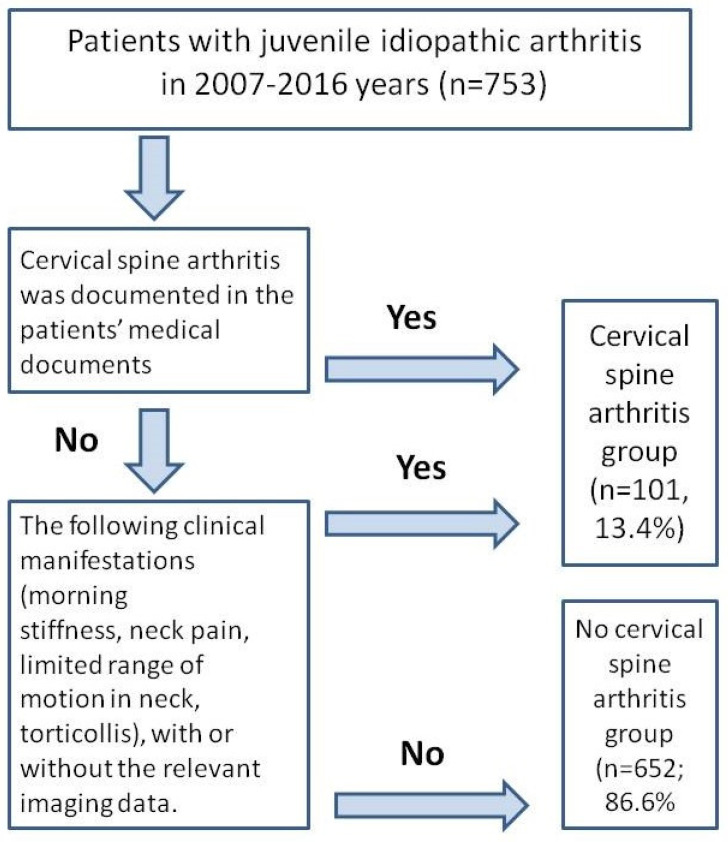
The flowchart of the study.

**Figure 2 jfmk-10-00068-f002:**
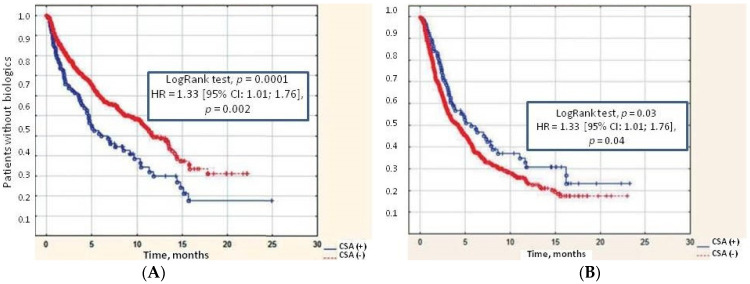
(**A**) Cumulative probability of biologic treatment requirement in patients with JIA with and without CSA; (**B**) cumulative probability to achieve remission in patients with JIA with and without CSA. Abbreviations: CSA (+)—the presence of cervical spine arthritis, CSA (−) no cervical spine arthritis in the patient.

**Figure 3 jfmk-10-00068-f003:**
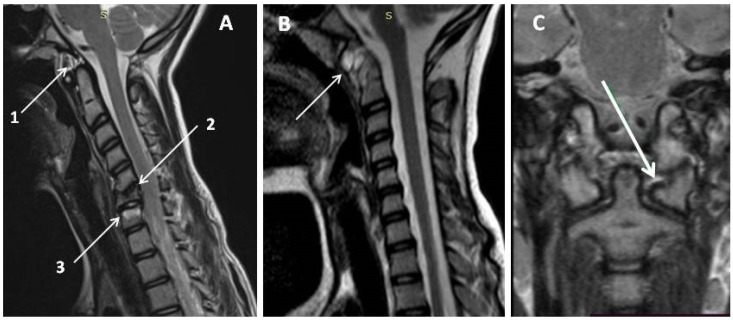
Synovitis of the atlantoaxial joint in a child with JIA 11 years old (own data): irregular width of the atlantoaxial joint (maximum up to ~8 mm); shows the accumulation of the heterogeneous fluid and proliferation of the synovium: (**A**,**B**)—sagittal (A1 and arrow in (**B**)); A2—postinflammatory changes, A3—cervical spondylodiscitis); (**C**)—frontal shows the inflammation of the atlanto-occipital joint, arrow—sagittal; Abbreviations: S—sagittal projection.

**Table 1 jfmk-10-00068-t001:** The main features of patients with JIA with and without CSA.

Parameters	CSA, Yes (n = 101)	CSA, No (n = 652)	*p*
**Demography**			
Girls, n (%)	69 (68.3)	388 (59.5)	0.092
Onset age, years, Me (25%; 75%)	5.3 (2.7; 10.1)	6.1 (3.0; 10.4)	0.241
Duration of disease, years, Me (25%; 75%)	5.9 (3.2; 9.4)	4.0 (1.8; 7.2)	0.0003
JIA category, n (%) Oligoarthritis Polyarthritis Psoriatic arthritis Enthesitis-associated arthritis Systemic arthritis	5 (5.0) 48 (48.0) 7 (7.0) 22 (21.8) 19 (18.9)	199 (30.5) 217 (33.3) 33 (5.1) 164 (25.2) 39 (6.0)	<0.001
Uveitis, n (%)	9/76 (11.9)	107/444 (24.1)	0.018
**Articular status**			
Active joints, Me (25%; 75%)	16.0 (9.0; 28.0)	5.0 (3.0; 10.0)	<0.001
Joint involvement: TMJ, n (%) Shoulder joint, n (%) Elbow joint, n (%) Wrist joint, n (%) Metacarpophalangeal joint, n (%) Proximal interphalangeal joint, n (%) Distal interphalangeal joint, n (%) Hip joint, n (%) Knee joint, n (%)	24 (23.7) 30 (29.7) 35 (34.6) 62 (61.4) 44 (43.6) 53 (52.5) 24 (23.8) 45 (44.6) 61 (60.4)	19 (2.9) 19 (2.9) 80 (12.2) 142 (21.8) 120 (18.4) 139 (21.3) 46 (7.1) 108 (16.6) 262 (40.2)	<0.001 <0.001 <0.001 <0.001 <0.001 <0.001 <0.001 <0.001 0.0001
**Laboratory**			
ANA-positivity, n (%)	22/57 (38.6)	190/403 (47.2)	0.226
Erythrocyte sedimentation rate, mm/h, Me (25%; 75%)	12.0 (5.0; 31.0)	7.0 (3.0; 18.0)	0.0006
C-reactive protein, mg/L, Me (25%; 75%)	3.9 (0.0; 20.0)	1.1 (0.0; 9.2)	0.002
**Treatment**			
Oral corticosteroids, n (%)	37/101 (36.7)	115/651 (17.7)	0.00001
Methylprednisolone pulse therapy, n (%)	33/100 (33.0)	102/650 (15.7)	0.00003
Methotrexate, n (%)	87/99 (87.9)	486/568 (85.6)	0.541
Biologics, n (%)	68 (67.3)	283 (43.4)	0.000007
**Outcomes**			
Remission of JIA, n (%)	57 (56.4)	428 (65.6)	0.072
Time to JIA remission, years, Me (25%; 75%)	2.9 (1.5–5.1)	2.2 (1.1–4.6)	0.046
Flares of JIA, n (%)	10 (9.9)	128 (19.7)	0.018

**Abbreviations:** ANA—antinuclear antibodies, JIA—juvenile idiopathic arthritis, Me—median, TMJ—temporomandibular joints.

**Table 2 jfmk-10-00068-t002:** The risk factors for CSA in children with JIA.

Risk Factors	Se	Sp	OR (95% CI)	*p*-Value
Oral corticosteroids	0.37	0.82	5.3 (3.1; 8.7)	<0.001
Methylprednisolone pulse therapy	0.33	0.84	2.7 (1.7; 4.2)	<0.001
Biologics	0.67	0.57	2.7 (1.7; 4.2)	<0.001
Polyarticular JIA category	0.48	0.67	1.8 (1.2; 2.7)	<0.001
Systemic JIA category	0.19	0.94	3.6 (2.0; 6.6)	<0.001
TMJ arthritis	0.24	0.97	10.4 (5.4; 19.8)	<0.001
Shoulder arthritis	0.3	0.97	14.1 (7.5; 26.3)	<0.001
Elbow arthritis	0.35	0.88	3.8 (2.4; 6.1)	<0.001
Wrist arthritis	0.61	0.78	5.7 (3.7; 8.9)	<0.001
Metacarpophalangeal arthritis	0.44	0.82	3.4 (2.2; 5.3)	<0.001
Proximal interphalangeal arthritis	0.53	0.79	4.1 (2.6; 6.3)	<0.001
Distal interphalangeal arthritis	0.24	0.93	4.1 (2.4; 7.1)	<0.001
Hip arthritis	0.45	0.83	4.1 (2.6; 6.3)	<0.001
Knee arthritis	0.55	0.6	1.8 (1.2; 2.7)	<0.001

**Abbreviations:** CI—confidence interval, JIA—juvenile idiopathic arthritis, Se—sensitivity, Sp—specificity, OR—odds ratio, TMJs—temporomandibular joints.

## Data Availability

Data available on request due to legal restrictions and ethical reasons.

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
