# Peer review of "The Features of Children with Juvenile Idiopathic Arthritis with Cervical Spine Involvement in the Data from a Retrospective Study Cohort"

_jfmk, 2025, doi:10.3390/jfmk10010068_

Round 1
Reviewer 1 Report
Comments and Suggestions for Authors
I thank the journal editor for the invitation and hope that the comments below will be helpful to the authors in the process of improving their work.
1. Identification of patients with arthritis of the cervical spine (ACS):
The introduction clarifies the difficulty of diagnosis and treatment of ACS, however, it is not clear from the methodology whether patients with ACS were already diagnosed or whether they were classified as such based on the data obtained. I recommend that the authors explicitly detail how ACS cases were identified in this retrospective analysis.
2. Risk factors for ACS:
Significant risk factors such as temporomandibular joint (TMJ) arthritis and polyarticular arthritis are mentioned in the results. I think it is important for the authors to expand this section:
1) Include an explanation of the methods used.
2) Describe in more detail how these factors were identified as predictive,
3) Provide a fuller discussion of the clinical relevance of these factors and how they could be applied in clinical practice for early diagnosis and treatment of ACS.
3. Discussion:
Although the discussion is comprehensive and contrasted with the referenced literature, at some points the direct connection to the results obtained in the study is lost. I recommend that the authors revise and restructure this section to keep the focus on the main findings of their research, making sure that the data presented are easily identifiable and contextualised in relation to other studies.
Author Response
Reviewer 1
Comments and Suggestions for Authors
I thank the journal editor for the invitation and hope that the comments below will be helpful to the authors in the process of improving their work.
Dear Reviewer!
Thank you for your kind evaluation of our manuscript. Our answers (A) on your queries (Q) are below. All changes in the manuscript highlighted with color.
Q1) Identification of patients with arthritis of the cervical spine (ACS): The introduction clarifies the difficulty of diagnosis and treatment of ACS, however, it is not clear from the methodology whether patients with ACS were already diagnosed or whether they were classified as such based on the data obtained. I recommend that the authors explicitly detail how ACS cases were identified in this retrospective analysis.
A1) Dear Reviewer! Thank you so much for this clarifying suggestion.
Patients were assigned to the group of CSA if the patient corresponded at least one criteria i) CSA was documented in the patients’ medical documents and/or ii) patient had the following clinical manifestations (morning stiffness, neck pain, limited range of motion in neck, torticollis), with or without the relevant imaging data.
This statement has been added in the Methods section
Q2) Risk factors for ACS:
Significant risk factors such as temporomandibular joint (TMJ) arthritis and polyarticular arthritis are mentioned in the results. I think it is important for the authors to expand this section:
1) Include an explanation of the methods used. 2) Describe in more detail how these factors were identified as predictive, 3) Provide a fuller discussion of the clinical relevance of these factors and how they could be applied in clinical practice for early diagnosis and treatment of ACS.
A2) Dear Reviewer!
The new subsection Assessments and Outcomes were added in the Methods with extended explanation. The results were structurised too with two sub0divisions and additional statistical explanation added to the second part of the Results (finding predictors of CSA). The discussion section, according to your recommendations has expanded.
Q3. Discussion:
Although the discussion is comprehensive and contrasted with the referenced literature, at some points the direct connection to the results obtained in the study is lost. I recommend that the authors revise and restructure this section to keep the focus on the main findings of their research, making sure that the data presented are easily identifiable and contextualised in relation to other studies.
A3. Dear Reviewer! The discussion has been updated and 10 new references added.
Dear Reviewer!
I hope manuscript has become better after your suggestions.
Thank you so much!
On behalf of the Authors,
Mikhail Kostik, MD, Ph.D., Professor
Reviewer 2 Report
Comments and Suggestions for Authors
Dear Editor, thank you for the opportunity to review the article entitled "The characteristics of children with juvenile idiopathic arthritis with spinal involvement: data from a retrospective cohort study". The article is interesting, but some points should be clarified or modified.
The entire article should be formatted with a number in the introduction, materials and methods, etc.
Introduction
Lines 44 and 45, the continuation of the paragraph was interrupted.
Lines 75 and 76 The aim of the study: is to......
Usually, the aim of the study is written at the end of the introduction, not highlighted as a subchapter or subitem. Only if requested by the journal, otherwise modify it.
I still suggest reviewing the review; some passages are confusing.
Methods
Please add a figure of experimental designer
Results
They are well described
Discussion
The discussion is satisfactory and really discusses the results found. But my suggestion would be to have an extensive discussion, not divided into subchapters or subitems. For original articles, it does not seem to me that it should be used in the way that was discussed.
The authors described the limitations of their study and were very ethical about this point.
Complement in detail the directions for future studies on the topic, based on the results found.
Format the bibliographic reference according to the newspaper template.
Overall, it is an interesting study; suggestions will improve the quality of the article.
Author Response
Dear Reviewer!
Thank you for your kind evaluation of our manuscript. Our answers (A) on your queries (Q) are below. All changes in the manuscript highlighted with color.
Introduction
Q1) Lines 44 and 45, the continuation of the paragraph was interrupted.
A1) Dear Reviewer! Fixed.
Q2) Lines 75 and 76 The aim of the study: is to......
Usually, the aim of the study is written at the end of the introduction, not highlighted as a subchapter or subitem. Only if requested by the journal, otherwise modify it.
I still suggest reviewing the review; some passages are confusing.
A2) Dear Reviewer! This part of the manuscript was re-written
Q3) Methods
Please add a figure of experimental designer
A3) Dear Reviewer! The flowchart added.
Q4) Results
They are well described
A4) Dear Reviewer! Thank you so much.
Q5) Discussion
The discussion is satisfactory and really discusses the results found. But my suggestion would be to have an extensive discussion, not divided into subchapters or subitems. For original articles, it does not seem to me that it should be used in the way that was discussed.
A5) Dear Reviewer! Thank you so much. The Subchapters deleted according to your suggestion.
Q6) The authors described the limitations of their study and were very ethical about this point.
A6) Dear Reviewer! Thank you so much.
Q7) Complement in detail the directions for future studies on the topic, based on the results found.
A7) Dear Reviewer! In respect to the Journal’s auditorium the statement “The future studies might be addressed to assess the functionality of the CS as a surrogate biomarker of disease activity and articular damage in patients with JIA and CSA” at the end of the discussion added.
Q8) Format the bibliographic reference according to the newspaper template.
A8) Dear Reviewer! The reference list will be formatted at the end of the reviewing process to avoid duplicative work.
Overall, it is an interesting study; suggestions will improve the quality of the article.
Dear Reviewer!
The manuscript has become better after your suggestions.
Thank you so much!
On behalf of the Authors,
Mikhail Kostik, MD, Ph.D., Professor
Round 2
Reviewer 2 Report
Comments and Suggestions for Authors
Dear Editor, the authors have responded to the questions and accepted the suggestions requested. I recommend accepting the article.